# Mu-Opioid Receptor 1 and C-Reactive Protein Single Nucleotide Polymorphisms as Biomarkers of Pain Intensity and Opioid Consumption

**DOI:** 10.3390/brainsci13121629

**Published:** 2023-11-24

**Authors:** Aleksander Turczynowicz, Piotr Jakubów, Karolina Niedźwiecka, Julia Kondracka, Weronika Pużyńska, Mariola Tałałaj, Tomasz Guszczyn, Paweł Grabala, Oksana Kowalczuk, Szymon Kocańda

**Affiliations:** 1Department of Anesthesiology and Intensive Care for Children and Adolescents with Postoperative and Pain Treatment Unit, Medical University of Bialystok, 15-274 Bialystok, Poland; piotr.jakubow@umb.edu.pl (P.J.); kondrackajulia@gmail.com (J.K.); mariola.talalaj@umb.edu.pl (M.T.); 2Department of Cardiosurgery, Medical University of Bialystok, 15-276 Bialystok, Poland; karolina.niedzwiecka@sd.umb.edu.pl (K.N.);; 3Department of Palliative Medicine, Medical University of Bialystok, 15-276 Bialystok, Poland; weronika.puzynska@umb.edu.pl; 4Department of Pediatric Orthopedics and Traumatology, Medical University of Bialystok, 15-274 Bialystok, Poland; tomasz.guszczyn@umb.edu.pl (T.G.); ortdziec@umb.edu.pl (P.G.); 5Department of Clinical Molecular Biology, Medical University of Bialystok, 15-269 Bialystok, Poland

**Keywords:** acute pain, postoperative pain, oprm1, SNP, single nucleotide polymorphism

## Abstract

Children constitute a special group in pain therapy. Single nucleotide polymorphisms that are associated with differences in postoperative, inflammatory pain perception and opioid requirement are the A118G SNP in the mu-opioid receptor 1 (OPRM1) gene and the rs1205 CRP. This study aimed to determine connection between OPRM1 and rs1205 CRP SNPs in pediatric patients postoperatively and pain intensity, the opioid dose needed to control pain after scoliosis correction, and other clinical aspects. Genotypes of rs1205 CRP and OPRM1 polymorphisms in a sample of 31 patients were specified, and statistical analysis was performed in terms of age, genotype frequency, pain assessment, sufentanil flow, post-anesthesia care unit stay, and the use of coanalgesics. The frequency of A/A and A/G genotypes in the OPRM1 gene was in line with 1000Genomes data for the European population. Patients from the AG group of OPRM1 SNP more frequently required coanalgesics for adequate pain control; however, it was of weak statistical significance. Other parameters measured in the study were not statistically significant in relation to OPRM1 and CRP polymorphisms. The effect of SNPs on postoperative pain management and opioid therapy in children was not confirmed by this study. An expansion of the study sample and other opioid-related SNPs is required.

## 1. Introduction

Research on the metabolomic diagnosis of pain is a rapidly advancing field in medicine. Pain is now considered not only as an indicator of treatment effectiveness but also may become a separate pathologic entity [1]. Children, due to their unique pharmacokinetic and pharmacodynamic characteristics, as well as cognitive and emotional immaturity, constitute a special group of patients when it comes to pain therapy [2]. These factors significantly influence the perception of pain and the efficacy of opioid therapy in pediatric patients, and pain management in pediatric settings is a large concern for physicians to this day, resulting often in inadequate analgesia provided for children experiencing pain [3,4,5].

Pediatric patients with scoliosis are at high risk of experiencing pain—both preoperatively and postoperatively. Long-term studies of unoperated idiopathic scoliosis in children indicate the increased progression of disability, including chronic pain [6].

The incidence of low back pain in children years after surgery range 7–77%, and back pain was reported in 61.5% of surgically treated children during a 4.8-year follow-up [7].

Other studies indicate less but also significant chronic pain after spine operation [8].

In addition to minimal invasive surgery, effective postoperative rehabilitation, individual psychological factors, and multimodal treatment, well-controlled pain after surgical trauma is a crucial factor of chronic pain prophylactic treatment [9,10,11].

Providing effective acute pain management to hospitalized children can help improve outcomes, decrease the length of stay, and increase patient and parental satisfaction [12].

Opioids are still the drugs of choice in postoperative pain treatment after surgery with a high level of tissue damage, such as scoliosis correction surgery, but they exhibit individual variability in their effectiveness across all age groups, and the previously mentioned factors create additional challenges in adequately managing acute pain in children, exposing this patient group to complications such as persistent postoperative pain [13,14,15]. In regard to the abovementioned issues, there are observations of opioid medication overprescription for postoperative pain, which can lead to nonmedical opioid misuse. That produces debates about necessity of accurate opioid administration in accordance with actual patients’ requirements to relieve pain [16].

As a result, one branch of pain research focuses on single nucleotide polymorphisms (SNPs) in acute and inflammatory pain as potential objective pain markers. SNPs can act as predictors of pain intensity and may indicate an increased need for opioid analgesics in the perioperative period [17,18]. One potential SNP associated with differences in pain perception and opioid requirements is the A118G SNP in the mu-opioid receptor 1 (OPRM1) gene. The mu-opioid receptor gene (OPRM1) codes for the mu-opioid receptor, which binds beta-endorphin. The A118G polymorphism in this gene affects beta-endorphin binding such that the Asp40 variant (G allele) binds beta-endorphin three times more tightly than the more common Asn40 variant (A allele) [19].

Another factor that may be considered is the SNP rs1205 in the C-reactive protein (CRP) gene affecting CRP plasma levels, which is a commonly used diagnostic marker of inflammation, which may also be connected to the severity of inflammatory pain caused by surgical interventions [20].

The assessment and treatment of pain in children based on their individual genetic predispositions is still a very little-discussed topic in scientific literature and clinical practice. Most of the available literature on the impact of the OPRM1 A118G and rs1205 CRP SNPs on acute and chronic pain largely concerns the population over 18 years of age. The aim of this article is to investigate whether the OPRM1 A118G and rs1205 CRP SNPs influence the perception of pain and the need for opioid analgesics immediately after orthopedic correction of scoliosis in a group of patients under 18 years of age.

## 2. Materials and Methods

### 2.1. Patient Population

The study was conducted at the Children’s Clinical Hospital in Bialystok (Poland), to which children from lower-reference centers from all over the country were referred. The selection was random, according to the queue of applications to the hospital.

A cohort study was conducted from May 2022 to April 2023, including all patients scheduled for surgical correction of scoliosis. The timeframe aligned with permission from the Bioethical Committee and funding acquisition from a scientific grant provided by the Medical University of Bialystok. 

The inclusion criteria were spinal scoliosis surgeries in children aged 0–18 years, an initial pediatric assessment, and a group I–II ASA anesthetic scale classification based on a thorough subjective and physical examination conducted in the presence of a legal guardian by an anesthesiologist, so they either had no health problems or any health problems were minor, well controlled, and stable.

The exclusion criteria were as follows: neurological disorders; patients with neuropathic pain associated with scoliosis requiring surgical treatment, including those who required opioid therapy in the period up to 3 months before the procedure to control pain; pulmonary disorders observed in spirometry, with particular emphasis on diseases causing pulmonary restriction, metabolic disorders, cardiac disorders including cardiac arrhythmias, congenital or acquired heart defects, psychiatric disorders—including borderline, histrionic, and schizoid personality disorders, phobias, anxiety, all types of depressive disorders, atypical bipolar disorder, or a suspicion of substance abuse (such patients were referred for evaluation by a psychologist and a psychiatrist). Possible symptoms of mental disorders were assessed during routine pediatric and anesthetic qualifications—if such symptoms were detected, the assessment was extended to include additional psychological and psychiatric consultations. If the patient was unfit for surgery, they were referred to other forms of treatment and disqualified from the study. 

The study was conducted following the Helsinki Declaration and was approved by the Bioethical Commission of the Medical University in Bialystok (approval no. APK.002.512.2021 from 16 December 2021)

### 2.2. Anesthesia and Postoperative Analgesia

Before arriving to the OR, the patients received oral midazolam at a dose of 0.2 mg/kg and, after the insertion of an intravenous line, a crystalloid infusion as a premedication. After the arrival of the patient at the OR, checking the identity, and the completion of the medical documentation, ECG, etCO_2_, blood saturation, the spectrophotometric measurement of oxygen in the respiratory mixture, and the gasometric measurement of critical condition parameters includingcomplete blood count and electrolytes were used, and the blood deficiency was supplemented according to the measurement of hemoglobin concentration, hematocrit, and estimated amount of blood lost.

Each patient was fitted with at least 2 additional wide-lumen peripheral catheters, a radial artery cannula to measure direct arterial blood pressure, BiSpectral Index (BIS) neuromonitoring to assess the depth of anesthesia, and peripheral potentials from the limbs and peripheral body parts to evaluate possible nerve damage caused by surgical factors.

The induction of anesthesia was performed through administering propofol and rocuronium according to body weight, and as an analgesic, an infusion of remifentanil was started; then, intubation was performed with an armored tube of a size adapted to age. Conduction was performed using total intravenous anesthesia target-controlled infusion pumps (TIVA-TCI) with a continuous infusion of propofol and remifentanil. The initial dose was calculated according to body weight, and then the infusion rate was adjusted according to BIS readings—maintaining values between 35 and 50, i.e., within the normal BIS values for TIVA general anesthesia, indicating an adequate depth of anesthesia.

Before the conclusion of the surgery and patient transport to the postoperative ward, a sufentanil infusion pump was added at a rate of 0.1 microgram/kg/h to continue analgesia on the post-anesthesia care unit (PACU) and prevent hyperalgesia upon stopping the remifentanil infusion. After surgery, all patients admitted to the PACU were administered a paracetamol infusion at a dose of 15 mg/kg and metamizole at a dose of 16 mg/kg as part of analgesia. Pain was assessed by nurses specialized in anesthesiology and intensive care, trained with an in-hospital procedure of pain management using the Visual Analogue Scale (VAS) for patients over 6 years old when they were able to verbally report subjective pain intensity, and the Face, Legs, Activity, Cry, Consolability (FLACC) scale for younger patients [2,13]. Parameters evaluated during the PACU stay included the sufentanil pump infusion rate at admission and the modified sufentanil pump infusion rate after the first pain assessment post extubation, at discharge from the PACU, and 24 h after surgery. This information was used to calculate the mean sufentanil pump rate (micrograms/kg/h) and pain intensity measured at four time points—post extubation, after the first infusion rate adjustment, at discharge, and 24 h post surgery. An additional parameter assessed was the C-reactive protein (CRP) level 72 h after surgery as an indicator of inflammatory pain. In cases of severe pain and high sufentanil doses for treatment, which led to the occurrence of severe side effects such as confusion, excessive sedation, and opioid-induced apnea and trouble breathing, sufentanil infusion was no longer increased, but co-analgesics were administered in the form of lidocaine at a rate of 1–1.5 mL/h and ketamine at a dose of 0.2–0.4 mg/kg. The use of lidocaine and ketamine as analgesics is justified from the point of view of clinical practice, which is supported by the Polish Society of Anaesthesiology and Intensive Care guidelines for the treatment of acute pain in children, as there is strong evidence of the effectiveness of lidocaine (IB) and ketamine (IA) as coanalgesics in acute pain [13].

### 2.3. Genotyping Data

Peripheral blood samples were collected to 1.2 mL EDTA test tubes from anesthetized patients before start of the surgery. Every sample was prepared through mixing blood with erythrocyte-lysis buffer, cooled down to 4 °C Celsius, and centrifuged for 10 min at 2500 rounds per minute to get an erythrocyte-free sediment of nucleated cells used for genetic testing. Genotyping was performed in the Department of Clinical Molecular Biology of Medical University of Bialystok on using Roche Diagnostics LightCycler*^®^* 480 (Roche Diagnostics International Ltd., Rotkreuz, Switzerland) real-time PCR hybridization probes for two sequence SNPs with ready-made TaqMan™ SNP Genotyping Assays rs1799971 (OPRM1 A118G) and rs1205 (CRP) from AppliedBiosystems™ (Foster City, CA, USA). Genotyping and post-genotyping analyses were conducted according to the assays’ manufacturer protocol. 

### 2.4. Statistical Analysis

The significance level of the statistical tests in this analysis was set at α = 0.05.

The normality of the distributions of the variables was analyzed using the Shapiro–Wilk test.

For numerical variables with distributions deviating from the normal distribution, the results were reported as the median (*Mdn*), along with the first quartile (*Q1*) and third quartile (*Q3*). For variables that followed a normal distribution, the results were reported as the mean (*M*) along with the standard deviation (*SD*).

For nominal or categorical variables, the frequency (n) and percentage (%) of each category were reported.

To assess the differences between two independent groups within to assess the differences between two independent groups within numerical variables characterized by a non-normal distribution, the Wilcoxon rank sum test (also known as the Mann–Whitney U test) was used. For variables involving more than two groups, the Kruskal–Wallis rank sum test was used.

For variables that followed a normal distribution, differences between more than two groups were assessed using the ANOVA Welch test. The significance of pairwise differences was estimated using the Dunn post hoc test with adjustment via the Holm method. 

The independence of two categorical variables was examined with Fisher’s exact test.

Analyses were conducted using the R Statistical language (version 4.1.1) [21] on Windows 10 Pro 64 Bit (build 19045), using the packages *report* (version 0.5.7) [22], *psych* (version 2.1.6) [23], *gtsummary* (version 1.6.2) [24], and *readxl* (version 1.3.1) [25].

## 3. Results

### 3.1. Characteristics of the Sample

Thirty-seven patients were enrolled in the study after obtaining signed informed consent. Due to the sampling error (n = 3) and missing samples due to laboratory error (n = 2), the final group included in the analysis comprised 31 patients. In the analyzed sample of N = 31 patients, there was a significant overrepresentation of females, accounting for 82.4% (n = 28) of the cohort, while males constituted a mere 17.6% (n = 6). The median age of the participants was 15 years, spanning an interquartile range (IQR) from 13.0 to 16.0 years, suggesting a relatively narrow age distribution around adolescence.

The largest proportion of subjects (52.9%, n = 18) fell within the 15–19-years category, followed by the 10–14-years group, accounting for 35.3% (n = 12). The under-9-years-of-age group was the least represented, consisting of just 11.8% (n = 4) of the total sample.

The anthropometric data showed a median BMI of 19.3 with an IQR from 17.2 to 21.6. This indicated that the majority of the cohort had a BMI falling within the parameters of a healthy weight, per the standards established by the World Health Organization.

### 3.2. Estimation of the Relationship between Selected Sociodemographic and Clinical Parameters of Patients and OPRM1 rs1799971 SNP

The results of comparing two genotypes (AA and AG) of the OPRM1 polymorphism in a sample of N = 31 patients in terms of age, age group, genotype frequency, pain assessment after extubation, drug flow in [µg/h] including per 1 kg weight, post-anesthesia care unit (PACU) stay, and the use of coanalgesics are shown in Table 1 and Figure 1 and Figure 2

The Pearson chi-square test presented statistically significant difference between AA and AG genotype frequency (*p* = 0.001). The median age for patients in the AA group was 15 years, while those in the AG group had a median age of 13 years. However, this difference was not statistically significant (*p* = 0.302), as determined by the Wilcoxon rank-sum test, suggesting that age did not differ significantly between these two genetic variants.

In terms of age groups, most patients with the AA variant (61.5%) were in the 15–19-year age group, whereas the majority of patients with the AG variant (60.0%) were in the 10–14-year age group. However, Fisher’s exact test revealed that these differences were not statistically significant (*p* = 0.224), indicating that the distribution of age groups was not significantly different between the AA and AG variants.

The pain assessment after extubation had a median score of 4.0 in the AA group and 5.0 in the AG group. The difference was not statistically significant (*p* = 0.684), as determined by the Wilcoxon rank-sum test, suggesting that the pain experienced after extubation did not differ meaningfully between these two genetic variants (Figure 3).

The means of both absolute and weight-adjusted drug flow rates in the case of the flow of the drug ([µg/h] and [µg/kg/h]) were slightly lower in the AG group compared to the AA group, but the *p*-values from the t-Welch tests (0.970 and 0.790, respectively) indicated that these differences are not statistically significant.

The median length of PACU stay was marginally shorter in the AG group compared to the AA group, but the *p*-value of 0.610 from the Wilcoxon rank sum test indicates that this difference is not statistically significant.

The proportion of patients who used coanalgesics was higher in the AG group compared to the AA group, but the *p*-value of 0.333 from Fisher’s exact test indicated that this difference was not statistically significant.

Based on the results of the statistical analyses, there were no statistically significant differences between the AA and AG groups in terms of patient age, pain assessment after extubation, the flow rate of the drug (both in absolute terms and relative to patient weight), the length of stay in the PACU, and the use of coanalgesics. Therefore, the genotype of the OPRM1 polymorphism (either AA or AG) did not appear to be associated with these patient characteristics and clinical parameters, as measured in this study.

### 3.3. Estimation of the Relationship between Selected Sociodemographic and Clinical Parameters of Patients and rs1205 CRP Polymorphism Types

The results of comparing three genotypes (CC, CT, and TT) of the rs1205 CRP polymorphism in a sample of N = 31 patients in terms of age, age group, genotype frequency, pain assessment after extubation, drug flow in [µg/h] including per 1 kg weight, post-anesthesia care unit stay, and the use of coanalgesics are shown in Table 2 and Figure 4.

Pearson chi-square test presented significant correlation among genotypes in rs1205 SNP with *p*-value of 0.001.

The median age differs between the three groups: 15 years for CC, 13.5 years for CT, and 16 years for TT. The Kruskal–Wallis rank sum test indicated a significant difference in age among these three genotypes (*p* = 0.021).

As for age groups, the majority of the patients with the CC genotype were in the 15–19-year group (60.0%), the majority of patients with the CT genotype were in the 10–14-year group (50.0%), and all patients with the TT genotype were in the 15–19-year group (100.0%). Despite these differences, Fisher’s exact test showed that the age group distribution was not significantly different among the three genotypes (*p* = 0.121).

The pain assessment after extubation presented mean scores of 4.2 (SD = 2.8) for the CC group, 5.3 (SD = 2.6) for the CT group, and 4.8 (SD = 2.2) for the TT group. The ANOVA Welch test indicated no significant difference in pain assessment scores among the three genotypes (*p* = 0.630).

The mean flow rates of the drug, both in absolute terms and relative to patient weight, do not differ significantly across the three groups (*p* = 0.900 and *p* = 0.470, ANOVA Welch tests, respectively).

The median length of PACU stay did not significantly differ across the three groups (*p* = 0.189, Kruskal–Wallis rank sum test).

The proportion of patients who used coanalgesics did not significantly differ across the three groups (*p* = 1.000, Fisher’s exact test).

There was no statistically significant difference observed in the pain assessment after extubation, the flow rate of the drug, the length of stay in the PACU, and the use of coanalgesics across the three groups. This suggested that the rs1205 CRP polymorphism genotype did not appear to have a significant impact on these clinical parameters and outcomes in the studied population.

### 3.4. Investigation of the Relationship between PACU Stay and Post-Extubation Pain Assessment, the Flow of the Drug, [ug/h] and Flow of the Drug, [µg/kg/h]

Table 3 presents the results of a Spearman correlation analysis between post-extubation pain assessment, the flow of the drug in [µg/h], the flow of the drug in [µg/kg/h] (further selected clinical parameters) and the length of stay in the PACU.

For post-extubation pain assessment, the very weak positive correlation (rho = 0.06) suggested that higher pain scores post extubation were associated with slightly longer PACU stays. In the case of the flow-of-the-drug parameter, the weak negative correlation (rho = −0.20) implied that higher drug flow rates might be associated with shorter PACU stays. For the flow of the drug in [µg/kg/h], the very weak negative correlation (rho = −0.08) indicated that when the drug flow rate was adjusted for patient weight, higher rates might be slightly associated with shorter PACU stays. However, none of these correlations reached statistical significance, as indicated by *p*-values exceeding the conventional 0.05 threshold. Thus, the correlations could likely be attributed to chance rather than reflecting true underlying relationships.

The analysis did not find evidence of a statistically significant correlation between the selected clinical parameters and the length of PACU stay. This suggested that these parameters, as measured in this study, may not be useful predictors of the duration of PACU stay.

### 3.5. Investigation of the Relationship between the Stay in the PACU and the Use of Coanalgesics

Table 4 presents the analysis of the relationship between the use of coanalgesics and the length of stay in the PACU.

From the results in Table 4, for patients who did not use coanalgesics, the median length of stay in the PACU was 2.0 h, with an interquartile range (IQR) from 1.8 to 2.5 h. On the other hand, for patients who did use coanalgesics, the median length of PACU stay was slightly longer, at 2.3 h, with an IQR from 2.1 to 2.5 h. However, the *p*-value from the Wilcoxon rank sum test was *p* = 0.304, which was greater than the conventional cutoff of 0.05 for statistical significance. Based on these results, although the median length of PACU stay was slightly longer for patients who used coanalgesics compared to those who did not, this difference was not statistically significant. Therefore, we cannot conclude that the use of coanalgesics affects the length of PACU stay based on this analysis.

## 4. Discussion

Uncontrolled postoperative pain results in persistent pain for many years after surgery [26,27]. Pain-reducing techniques include atraumatic surgical procedures, new perioperative techniques such as ERAS, and multimodal pain management, including the use of distraction methods, anxiolytic effects, sleep improvement, and antipsychotic effects [28,29]. One of the elements that can be used in parallel with the above techniques is to determine the pain predisposition of patients through determining the genotype that shows reduced sensitivity to opioids.

According to the literature, genetic polymorphism can have a significant impact on the level of perceived pain [18]. This is related to individual variability, different sensitivity to painful stimuli, and varying drug metabolism, which is especially concerning for clinicians regarding pain assessment and adequate treatment administration in pediatric patients [3,30]. It has been shown in the literature that the presence of the G allele in A118G OPRM1 SNP is associated with higher opioid requirement and subsequently—individual pain perception [31]. In meta-analyses from 2014 of 4607 patients and 2019 of 8609 patients considering A118G SNP effect on opioid dose interindividual variability has shown that G-allele carriers required a higher opioid dose to achieve satisfactory pain control [32,33]. However, some authors do not confirm such a relationship and demonstrate that the level of perceived pain is not directly dependent on gene polymorphism [34,35]. 

This variability in the significance of the OPRM1 A118G SNP may be dependent on patient ancestry, as the G allele is more common in patients of Asian or African descent than in European ones [36,37]. In study regarding a large group of Scandinavian patients undergoing cancer pain treatment with opioids, no connection between OPRM1 polymorphism and pain or opioid demand was found [38]. In recent studies, researchers explored the possibility of combining the A118G SNP with other SNPs, such as catechol-o-methyltransferase (COMT) or ATP-binding-cassette protein 1 (ABCB1) polymorphisms, which presented that interactions of SNPs may play role in indicating high-pain-risk patients as well as higher opioid requirements after surgery [39,40].

In the literature, CRP and its related SNP rs1205 are not commonly related as sources of acute inflammatory reaction and connected to surgically induced inflammation and pain; however, we found one study which addresses this matter. The SNP used in our study is used as a part of the haplotype of the CRP gene, which presented higher CRP levels after cardiac procedures [41].

In the present study, we found that the frequency of A/A and A/G genotypes in the OPRM1 gene was in line with 1000Genomes data for the European-ancestry population [36]. 

According to the pain assessment, we observed higher pain levels after extubation in patients in the A/G group, but these were not statistically significant. We also observed more frequent need for co-analgesics in patients with the A/G genotype. No statistically significant differences were found between the A/A and A/G groups regarding drug infusion rates (both in absolute values and in relation to patient body weight), the length of stay in the postoperative care unit, and the use of co-analgesics. Therefore, it does not seem that the OPRM1 genotype (A/A or A/G) is associated with the parameters studied.

This is consistent with data in the literature on perioperative studies in children, where it was shown that SNPs influence the level of pain and the response to analgesics [37].

In pain control, it is important to adopt a comprehensive, multimodal approach using many techniques, including behavioral distraction [42]. Therefore, joint treatment consisting of the assessment of genetic predisposition and individual mental constitution as well as determining the behavioral response to pain provides a chance to improve pain treatment in children and reduces the risk of chronic pain and potential treatment-related complications, including addiction [43]. 

In recent years, the possibility of SNP determination in the form of rapid diagnostic tests has been introduced [44]. The potential use of SNPs as rapid diagnostic tests may contribute to their use in clinical practice.

Despite the inability to prove the effect of SNPs on pain dose and intensity in our study, existing scientific evidence provides hope that this method may be used in the future to adjust drug doses and pain control methods once stronger evidence is obtained from large-scale studies on larger patient populations.

## 5. Conclusions

This study is the first to assess the possibility of using the OPRM1 A118G SNP in postoperative pain management in children undergoing major spinal fusion surgery, which obviously is an indication for postoperative opioid therapy. Due to the selection of only patients undergoing only one type of operation, this study is not biased by patients who experienced different surgical stimuli, and only patients of Central European ancestry were studied, which could produce different results. 

The strength of our study is in addressing the very important matter of pain management in children, which is not as widely discussed in the literature as in adults. This study did not find any significant differences between different genotypes of OPRM1 and CRP genes. We found that patients among the AG genotype group of the OPRM1 SNP required coanalgesics more frequently than patients in the AA group (80% vs. 46.2%). However, statistical analysis using Fisher’s exact test presented a very weak level of significance. 

A weakness of the study is the unsatisfactory sample size, as other studies regarding OPRM1 were often performed on well over 100–200 patients, coming close to even 900 [18,31,37]. As previously mentioned, there are other SNPs which have been widely researched for their contribution to opioid metabolism and pain perception, such as COMT rs4680, rs4633, rs4818, rs6269, and ABCB1, which were not investigated in this study [45,46]. 

In a small group of studied patients, the thesis about the influence of nucleotide polymorphisms of a single gene on the perception of pain was not confirmed. We believe that the key to effective pain control in children is still the individual variability of the child’s behavior, as well as accurate pain assessment in the PACU and the individual adjustment of pain medication.

This study will be continued to expand the study sample and include genetic data from other opioid-related SNPs to provide more qualitative data on the feasibility of tailored pain management in children with single nucleotide polymorphisms. 

## Figures and Tables

**Figure 1 brainsci-13-01629-f001:**
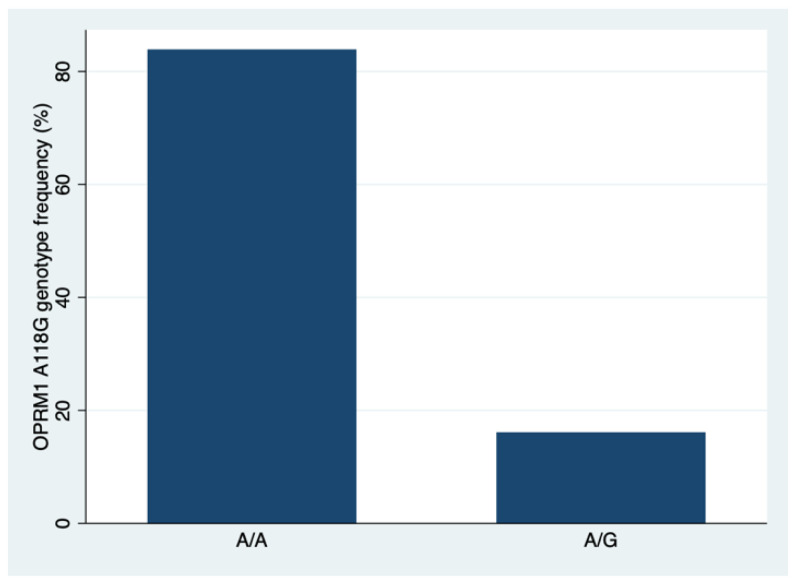
A/A genotype is more frequently present than A/G genotype among studied sample of Central Europeans (*p* = 0.001).

**Figure 2 brainsci-13-01629-f002:**
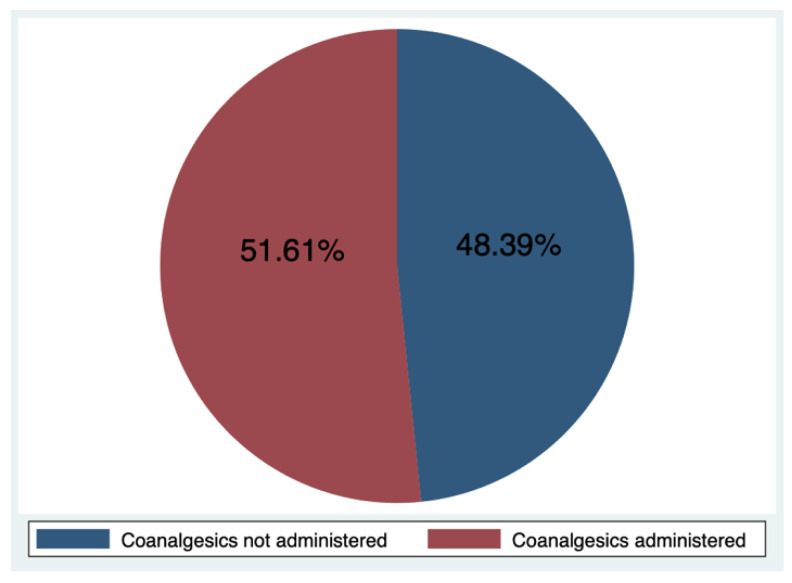
Coanalgesics administration frequency (%).

**Figure 3 brainsci-13-01629-f003:**
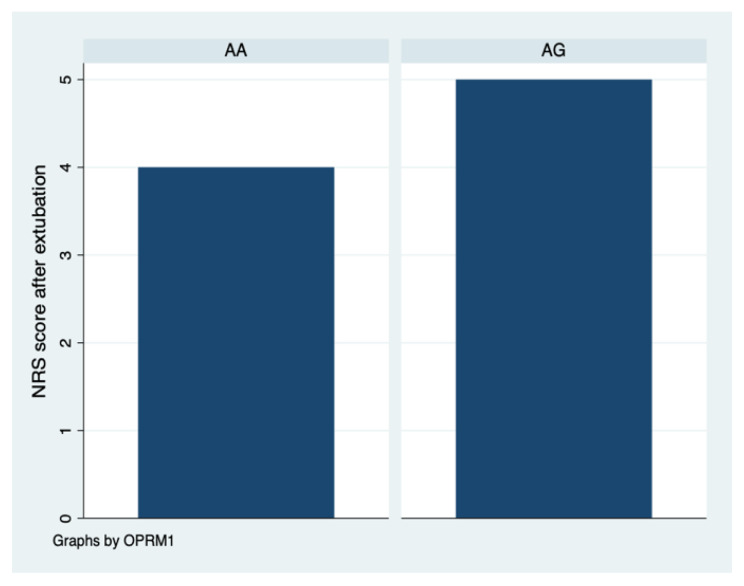
Pain intensity after extubation is higher in A/G than A/A genotype.

**Figure 4 brainsci-13-01629-f004:**
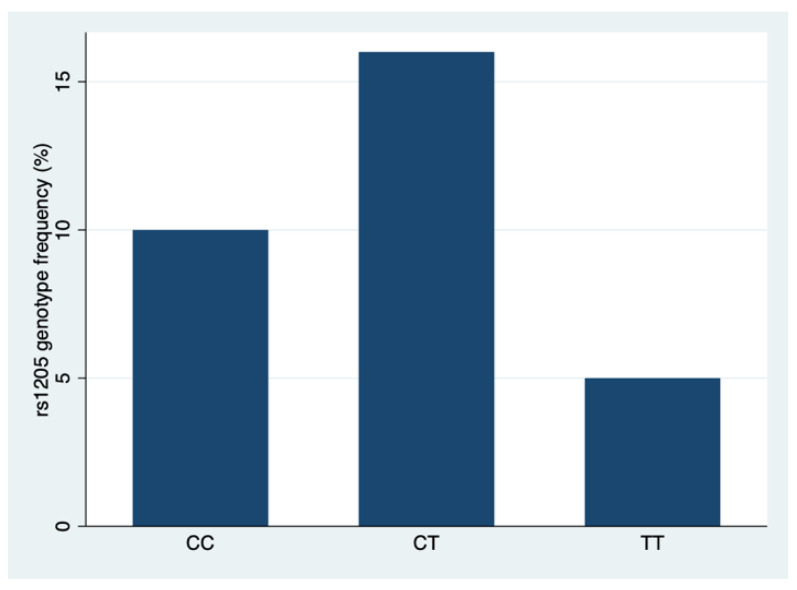
rs1205 SNP genotype frequencies (%).

**Table 1 brainsci-13-01629-t001:** Parameter distributions of patient age and pain assessment by genotypes of OPRM1 polymorphism with an estimation of the significance of differences between groups.

Characteristic	N	OPRM1		*p* ^3^ 0.001
		AA, n = 26 (83.9%) ^1^	AG, n = 5 (16.1%) ^1^	
Age, years	31	15.0 (13.0, 16.0)	13.0 (11.0, 14.0)	0.302
Age group:	31			0.224 ^4^
up to 9 yrs.		3.0 (11.5%) ^2^	1.0 (20.0%) ^2^	
10–14 yrs.		7.0 (26.9%) ^2^	3.0 (60.0%) ^2^	
15–19 yrs.		16.0 (61.5%) ^2^	1.0 (20.0%) ^2^	
Pain assessment after extubation	31	4.0 (2.0, 7.0)	5.0 (4.0, 6.0)	0.684
Flow of the drug, [µg/h]	31	4.2 (1.6) ^5^	4.1 (2.0) ^5^	0.970 ^6^
Flow of the drug, [µg/kg/h]	31	0.09 (0.04) ^5^	0.09 (0.01) ^5^	0.790 ^6^
Stay in the PACU, hours	31	2.2 (1.9, 2.5)	2.1 (2.0, 2.3)	0.610
Use of coanalgesics:	31			0.333
no		14.0 (53.8%)	1.0 (20.0%)	
Yes		12.0 (46.2%)	4.0 (80.0%)	

^1^ Mdn (Q1, Q3); ^2^ n (%); ^3^ Wilcoxon rank sum test; ^4^ Fisher’s exact test; ^5^ M (SD); ^6^
*t*-Welch test.

**Table 2 brainsci-13-01629-t002:** Parameter distributions of patient age and pain assessment by genotypes of rs1205 CRP polymorphism with an estimation of the significance of differences between groups.

Characteristic	N	rs1205 CRP			*p* ^4^ 0.001
		CC, n = 10 (32.3%) ^1^	CT, n = 16 (51.6%) ^1^	TT, n = 5 (16.1%) ^1^	
Age, years	31	15.0 (12.2, 15.8)	13.5 (11.8, 15.0)	16.0 (16.0, 16.0)	0.021
Age group:	31				0.121 ^5^
up to 9 yrs		2.0 (20.0%) ^2^	2.0 (12.5%) ^2^	0.0 (0.0%) ^2^	
10–14 yrs		2.0 (20.0%) ^2^	8.0 (50.0%) ^2^	0.0 (0.0%) ^2^	
15–19 yrs		6.0 (60.0%) ^2^	6.0 (37.5%) ^2^	5.0 (100.0%) ^2^	
Pain assessment after extubation	31	4.2 (2.8) ^3^	5.3 (2.6) ^3^	4.8 (2.2) ^3^	0.630 ^6^
Flow of the drug, [µg/h]	31	4.03 (1.88) ^3^	4.29 (1.65) ^3^	4.0 (1.25) ^3^	0.900 ^6^
Flow of the drug, [µg/kg/h]	31	0.09 (0.04) ^3^	0.09 (0.03) ^3^	0.07 (0.03) ^3^	0.470 ^6^
Stay in the PACU, hours	31	2.2 (1.9, 2.7)	2.0 (1.8, 2.3)	2.4 (2.4, 2.5)	0.189
Use of coanalgesics:	31				1.000
no		5.0 (50.0%)	8.0 (50.0%)	2.0 (40.0%)	
yes		5.0 (50.0%)	8.0 (50.0%)	3.0 (60.0%)	

^1^ Mdn (Q1, Q3); ^2^ n (%); ^3^ M (SD); ^4^ Kruskal–Wallis rank sum test; ^5^ Fisher’s exact test; ^6^ ANOVA Welch test.

**Table 3 brainsci-13-01629-t003:** Results of correlation analysis between selected clinical parameters and PACU stay.

Characteristic	Rho	*p*
Post-extubation pain assessment	0.06	0.732
Flow of the drug, [µg/h]	−0.20	0.250
Flow of the drug, [µg/kg/h]	−0.08	0.621

**Table 4 brainsci-13-01629-t004:** The distribution of length of stay in the intensive care unit grouped by the use of coanalgesics with estimation of the differences between the groups.

Characteristic	N	Use of Coanalgesics		*p* ^2^
		No, n = 15 ^1^	Yes, n = 16 ^1^	
Stay in the PACU, hours	31	2.0 (1.8, 2.5)	2.3 (2.1, 2.5)	0.304

^1^ Mdn (Q1, Q3); ^2^ Wilcoxon rank sum test.

## Data Availability

The data presented in this study are available on request from the corresponding author. The data are not publicly available due to privacy or ethics.

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
