# Peer review of "Mu-Opioid Receptor 1 and C-Reactive Protein Single Nucleotide Polymorphisms as Biomarkers of Pain Intensity and Opioid Consumption"

_brainsci, 2023, doi:10.3390/brainsci13121629_

Round 1
Reviewer 1 Report
Comments and Suggestions for Authors
The text discusses a study investigating the impact of genetic polymorphisms, particularly the A118G OPRM1 SNP and CRP rs1205, on the management of postoperative pain in pediatric patients, with a focus on spinal surgery in Central European children. The discussion explores the influence of genetic variability on pain perception and the necessity for opioid usage. The study references prior research and potential interactions among various genetic factors. However, the findings reveal no significant differences between genotypes concerning pain intensity, analgesic drug requirements, or clinical parameters. The study emphasizes the importance of personalized pain management and calls for further research with larger sample sizes and additional genetic data. Nonetheless, the article has room for improvement in several aspects.
- The introduction discusses various factors related to pediatric pain management but lacks a clear definition of the research gap or question the study aims to address. It is crucial to clearly state the research question or hypothesis, explaining what remains unknown or unexplored in the field and how the study intends to fill that gap.
- While the study's duration is provided (May 2022 to April 2023), the specific site where the study was conducted is not explicitly mentioned. It's essential to give information on the site, such as the hospital or medical institution where the research occurred and their routine procedures. Additionally, consider explaining why the study was conducted over this specific period (May 2022 to April 2023). Did this timeframe align with a particular need or research objective?
- In the inclusion criteria, it is mentioned that patients needed to be in overall good health for surgery. However, it would be beneficial to define what you mean by "overall health" or list specific health criteria that were considered.
- Clarify the reasoning behind excluding patients with prior opioid treatment. Did you anticipate that previous opioid treatment might influence the study results? Please elaborate on this. Similarly, while the exclusion of patients with chronic pain before surgery is reasonable, you could provide more context on what conditions were considered as chronic pain.
- Please explain how anxiety and depressive symptoms were assessed and whether standardized tools or criteria were used to identify these conditions.
- Provide more information about the anesthesia protocol, such as the doses of propofol and remifentanil used and any relevant monitoring parameters.
- Clarify how pain was assessed using the Visual Analogue Scale (VAS) and the Face, Legs, Activity, Cry, Consolability (FLACC) scale. Include details about who conducted these assessments and at what time points. Specify who administered and scored these scales.
- Explain the rationale behind using lidocaine and ketamine as co-analgesics in cases of severe pain. What specific criteria were used to determine when these co-analgesics were administered?
- While you mention the genotyping of OPRM1 and CRP, provide additional details on the laboratory methods used for DNA extraction and genotyping. This part can include specifics on equipment, protocols, and quality control measures.
- Be more explicit about the specific variables tested with each statistical method (Wilcoxon rank sum test, Kruskal-Wallis rank sum test, ANOVA Welch test, Fisher's exact test). Also, provide a brief rationale for the choice of each statistical test for the respective data.
- The discussion section should be improved and rewritten. It currently appears as a repetition of the findings. The findings should be compared with the results of previous studies in more detail, and information about the possible clinical implications of the findings should be added and discussed.
Author Response
- Long-term studies of unoperated idiopathic scoliosis in children indicate increased progression of disability, including chronic pain. The incidence of low back pain in children years after surgery ranges from 7-77%, and Back pain was reported in 61.5% of surgically treated children and 4.8-year follow-up. Other studies indicate less but also significant chronic pain after spine surgery.
In addition to effective postoperative rehabilitation, individual psychological factors, and multimodal treatment, uncontrolled pain after surgical trauma is an important predictor of chronic pain.
Providing effective acute pain management to hospitalized children can help improve outcomes, decrease length of stay, and increase patient and parental satisfaction.
Assessment and treatment of pain in children based on their genetic predispositions is still a topic very little discussed in scientific literature and clinical practice. Most of the available literature on the impact of the OPRM1 A118G and rs1205 CRP SNPs on acute and chronic pain largely concerns the population over 18 years of age. This article aims to investigate whether the OPRM1 A118G and rs1205 CRP SNPs influence the perception of pain and the need for opioid analgesics immediately after orthopedic correction of scoliosis in a group of patients under 18 years of age.
- The study was conducted at the Children's Clinical Hospital in Białystok (Poland), to which children from lower-reference centers from all over the country were referred. The selection was random, according to the queue of applications to the hospital.
The timeframe of the study was limited by the university's work schedule, including the limitation of scientific grant funding and the consent of the Bioethics Committee.
However, the study will continue to recruit a larger study group and use a larger number of SNPs related to opioid metabolism and pain perception.
- Inclusion criteria were spinal scoliosis surgeries in children aged 0-18 years, based on a thorough subjective and physical examination conducted in the presence of a legal guardian, were assessed on the ASA anesthetic scale as group I-II, so they had no health problems or any health problems were light. and well controlled, and stable.
The exclusion criteria were: neurological defects, patients with neuropathic pain associated with scoliosis requiring surgical treatment, including those who required opioid therapy in the period up to 3 months before the procedure to control pain, pulmonary disorders observed in spirometry, with particular emphasis on diseases causing pulmonary restriction, metabolic disorders, cardiac disorders - cardiac arrhythmias, congenital or acquired heart defects, psychiatric disorders including borderline, histrionic, schizoid personality disorders, phobias, anxiety all types depressive disorders or atypical bipolar were referred for evaluation by a psychologist and possibly to a psychiatrist and were referred to other forms of treatment.
- Previous opioid treatment for chronic pain may result in the need to use higher doses of opioids, which, like chronic pain, may have impaired the final results of the study, and one of the pain assessment criteria was the total dose of opioids administered in the PACU.
Patients with neuropathic pain related to scoliosis requiring surgery, including those who required preoperative opioid therapy to control pain, were disqualified.
- Possible symptoms of depression or anxiety disorders were assessed during routine pediatric and anesthetic qualifications - if such symptoms were detected, the assessment was extended to include additional psychological and psychiatric consultations.
- Before arriving at the OR, the patients received oral midazolam at a dose of 0.2 mg/kg and, after insertion of an intravenous line, a crystalloid infusion as a premedication. After arrival of the patient at the OR, checking the identity and compliance of the patient's medical documentation, ECG, etCO2, blood saturation, spectrophotometric measurement of oxygen in the respiratory mixture, gasometric measurement of critical condition parameters - complete blood count, electrolytes were used, and the blood deficiency was supplemented according to the measurement of hemoglobin concentration, hematocrit and estimated amount of blood lost.
Each patient was fitted with at least 2 additional wide-lumen peripheral catheters, a radial artery cannula to measure direct blood pressure, BiSpectral Index (BIS) neuromonitoring to assess the depth of anesthesia, and peripheral potentials from the limbs and peripheral body parts to evaluate possible nerve damage caused by surgical factors.
Induction of anesthesia was performed by administering propofol and rocuronium according to body weight, as an analgesic, an infusion of remifentanil was started, and then intubation was carried out with an armored tube of an age-appropriate size. During conduction, anesthesia was used in the TIVA protocol using a continuous infusion of propofol and remifentanil, the initial dose was calculated according to body weight, and then the infusion rate was adjusted according to BIS recommendations - maintaining the values between 35-50, i.e. within the values normal for TIVA general anesthesia, indicating good depth of anesthesia.
- Pain scales and treatment were administered by a PACU physician, but assessments were performed by PACU nurses specializing in anesthesiology and critical care and trained in the hospital procedure for pain assessment and sedation. Pain was assessed on a 100 mm VAS scale based on the subjective assessment of the patient over 6 years of age. The FLACC scale was used in children under 6 years of age. The pain was assessed four times by nurses - 15 minutes after extubation, after possible correction of the sufentanil dose, at discharge, and 24 hours after the procedure - by the nursing staff of the orthopedic clinic. This is detailed in the materials and methods section on postoperative anesthesia and analgesia.
- The use of lidocaine and ketamine as analgesics is justified from the point of view of clinical practice, which is supported by the PTAiIT guidelines for the treatment of acute pain in children - there is strong evidence of the effectiveness of lidocaine (IB) and ketamine (IA) as coanalgesics in acute pain. They were started by the PACU doctor when the dose of sufentanil was not sufficient to control the pain and the patient began to present side effects - excessive sedation, apnea, breathing difficulties.
- Specific data on genotyping process has been added to the methods section of the manuscript
- Please look at the upper indices in the p column. These indices indicate the type of test used, and at the bottom, there are explanations of which index refers to which test. By default (if there is no index above the p-value), the test with index 3 (Fisher exact test) was used.
The use of the above-mentioned tests depended on: the type of variable (numeric or nominal), the distribution of the numerical variable (normal or non-normal), and the number of compared groups (two or more).
the justification for using the tests is described in the statistical analysis methodology section.
- Thank you for your valuable commentary, the discussion has been edited and the study's potential implications for clinical practice has been added.
Reviewer 2 Report
Comments and Suggestions for Authors
On face value the study has great professionalism but having spent 40 years studying polymorphisms associated with addiction I have found this type of genetic data alone not medically helpful enough, without combination with a brain health check up or a Neuropsychiatric computer assessment ( ie cognitive attention mood sleep etc. ) since almost all people complaining of pain suffer phantom pain to 100% and of course less! Pain markers are rarely measured andESR and CRP are almost always negative as the T-helper suppress ratio usually shows excess suppressor cells relative to helper cells , but the more important things are left off this paper ie as above sleep quality of sleep, borderline personality, histrionic, schizoid social phobia , anxiety disorders of all types depressive disorders, atypical bipolar all of which completely dominate the pain spectrum with Marihuana and narcotics as self medication ! In addition poor utilization of benzodiazepines is common and under treatment of lots of alcoholism and drug abuse. Paper may be a value if some of these caveats are added to the paper ! the pediatrician to the future simply have to use genetic material and weigh it next to neuropsychiatric problems and comorbidities - Charleson comorbidities scores may be of help as a caveat to all pain Research in the older patients
Author Response
Thank you very much for your valuable comments.
The study was carried out in children referred for surgical correction of scoliosis of the spine at the University Hospital in Białystok, Poland. The study concerned only children operated on due to orthopedic indications for surgical treatment of spinal scoliosis. Children from reference centers from all over Poland were referred to the hospital center for orthopedic spine surgery.
The study assessed acute pain, i.e. postoperative pain in the immediate period after surgery in the postoperative ward. The depth of sleep-sedation was also assessed as an element of anesthetic treatment in the operating room through the BiSpectral Index as monitoring the depth of anesthesia and in the postoperative ward - the assessment of the 10-point sleepiness scale (Aldret) was performed by nurses specializing in anesthesiology, who were trained to assess postoperative sleep. and pain in children. Nurses assessing pain have undergone training confirmed by accreditation obtained by the hospital regarding postoperative management of acute pain. Assessment of postoperative pain and sedation was a standard procedure included as an identified procedure in hospital accreditation guidelines.
During the preparatory proceedings during an anesthetic visit among children referred for spine surgery, children with persistent pain and chronic pain diagnosed by the IASP recommendations as pain occurring for more than 3 months were excluded from the study.
In addition, after orthopedic qualification, all children underwent a standard pediatric qualification taking into account somatic disorders, and all children also underwent qualification and a possible psychological assessment in the event of detection of mental disorders to understand the extent and consent to spine surgery. The standard mode of preparation for surgery in the examined department, in addition to orthopedic and anesthetic examination and analysis of somatic co-morbidity, also includes psychological and possibly psychiatric examination in cases of identified necessity.
After this consultation, two children underwent psychiatric consultation, which excluded them from surgery. These children were referred to a different treatment mode and are currently under the care of a mental health clinic and have not been operated on. If mental disorders were detected by screening and, after subsequent psychological and psychiatric consultation, were assessed as severe, such patients were disqualified from our examination and the procedure by the orthopedist until the patient's health condition was stabilized.
The studied population was pediatric, and a possible diagnosis of the personality disorders mentioned above can be diagnosed only after the patient reaches the age of majority, because according to the current state of knowledge, human personality is formed even up to the age of 20-25, but despite the impossibility of making a diagnosis according to the DSM criteria V features of personality disorders or predisposing to their development in later life may be observed in children, which is certainly a factor worth considering in the clinical assessment of a patient qualified for extensive orthopedic surgery.
The assessment of the use of stimulants - alcohol, tobacco, and drugs - was performed during the pediatric visit and the anesthetic qualification for anesthesia. In case of suspected substance abuse, patients were disqualified from anesthesia and surgery after psychological and psychiatric consultation, as previous drug or alcohol abuse could significantly disturb the test results, increasing the need for anesthetics and analgesics.
If there was a suspicion of serious mental disorders or the patient reported such treatment during a pediatric or anesthesiological visit, he or she was also consulted psychologically and psychiatrically, and if serious disorders were found, he or she was disqualified from the study. Urine drug screening tests were randomly performed on children.
During the examination immediately after surgery, the children were under the influence of anxiolytic drugs and the residual effects of anesthetic agents, as well as the use of benzodiazepines in the preoperative period as a premedication before surgery. In this case, a short-acting benzodiazepine - midazolam was used, due to the standard treatment adopted in the center for all patients. The depth of sedation was assessed using the Ramsay scale, which is the standard of care in the postoperative department.
Children could leave the postoperative ward if they had a Ramsay score of 1-2 (fully conscious), which is consistent with the recommendations for discharge from the postoperative ward.
Chronic pain was not assessed and children with chronic pain as well as children with such pain were not qualified for the study.
Children were assessed for acute pain that occurred during recovery from anesthesia in the postoperative ward nym. In the immediate 24 hours after surgery.
The study did not examine addiction or potential psychological predispositions. The study focused on the assessment of pain, nociception by nurses, and the perception of pain reported by children after surgery.
During recovery from anesthesia after extensive surgery, neuropsychological assessment of mood, attention, and other cognitive functions is not possible for children within 3 hours. After waking up from anesthesia and extubation, children are sleepy, do not answer questions fully logically, and are under the end-stage effects of anesthetic drugs, but they already report pain. This involves 100% certainty that the assessment of cognitive functions will be impaired in this situation. By the recommendation of the group for the assessment of cognitive disorders after surgery, such tests, including the MoCA test, and the drug and depression scale, were performed more than 7 days after the surgery and they concern a different study. In this study, we assessed the condition of children after surgery, including acute pain and the depth of sedation as part of the management in the postoperative department.
We fully agree with the statement that genetic assessment itself does not replace a holistic assessment of the patient in terms of the symptoms reported by him or her, physical examination - it may only constitute a fragment of the clinical context in a given case, however, the study aimed to assess the impact of the studied SNPs on the intensity of pain and the need for opioids, which, as we know from clinical practice, can be highly variable from person to person, which causes difficulties for clinicians in selecting the drug and dose. Such an objective factor, such as SNPs in genes related to opioid metabolism, would allow the recognition of groups at high risk of pain, which would facilitate the selection of treatment and prevent many complications of opioid treatment, including the previously mentioned opioid addiction, which is currently a major health problem in many countries. places in the world.